# Inference of Regulatory System for TAG Biosynthesis in *Lipomyces starkeyi*

**DOI:** 10.3390/bioengineering7040148

**Published:** 2020-11-19

**Authors:** Sachiyo Aburatani, Koji Ishiya, Toshikazu Itoh, Toshihiro Hayashi, Takeaki Taniguchi, Hiroaki Takaku

**Affiliations:** 1Computational Bio Big-Data Open Innovation Laboratory (CBBD-OIL), National Institute of Advanced Industrial Science and Technology (AIST), Tokyo 305-8568, Japan; 2BPRI, National Institute of Advance Industrial Science and Technology (AIST), Sapporo 062-8517, Japan; koji.ishiya@aist.go.jp; 3Mitsubishi Research Institute, Inc., Chiyoda District, Tokyo 100-8141, Japan; ito@mri.co.jp (T.I.); toshihiro_hayashi@mri.co.jp (T.H.); tany@mri.co.jp (T.T.); 4Department of Applied Life Sciences, Niigata University of Pharmacy and Applied Life Sciences, Niigata 956-8603, Japan

**Keywords:** oleaginous yeast, network modeling, TAG biosynthesis, regulatory system

## Abstract

Improving the bioproduction ability of efficient host microorganisms is a central aim in bioengineering. To control biosynthesis in living cells, the regulatory system of the whole biosynthetic pathway should be clearly understood. In this study, we applied our network modeling method to infer the regulatory system for triacylglyceride (TAG) biosynthesis in *Lipomyces starkeyi*, using factor analyses and structural equation modeling to construct a regulatory network model. By factor analysis, we classified 89 TAG biosynthesis-related genes into nine groups, which were considered different regulatory sub-systems. We constructed two different types of regulatory models. One is the regulatory model for oil productivity, and the other is the whole regulatory model for TAG biosynthesis. From the inferred oil productivity regulatory model, the well characterized genes DGA1 and ACL1 were detected as regulatory factors. Furthermore, we also found unknown feedback controls in oil productivity regulation. These regulation models suggest that the regulatory factor induction targets should be selected carefully. Within the whole regulatory model of TAG biosynthesis, some genes were detected as not related to TAG biosynthesis regulation. Using network modeling, we reveal that the regulatory system is helpful for the new era of bioengineering.

## 1. Introduction

The improvement of the bioproduction ability in microorganisms is one of the important themes in bioengineering fields. Several types of empirical breeding approaches, such as constructing random mutant strains [1,2,3] and improving key enzyme activities within the biosynthesis pathways [4,5], have been developed over the years and applied to expand the capabilities of microorganisms. Those approaches have yielded useful host strains, but their development is quite time-consuming and costly. To solve this high cost problem, system approaches combined with usual breeding methods have been applied to produce useful host strains of microorganisms. Among the several host microorganisms used in bioproduction, the oleaginous yeast *Lipomyces starkeyi* is quite important. This produces edible oil with a fatty acid composition similar to that of palm oils, at a reasonable cost [6,7,8,9]. Thus, improving the oil production by this oleaginous yeast is a fascinating goal, for not only bioengineering but also in industrial use as an alternative to palm oil.

The specific feature of *L. starkeyi* is its oil accumulation system [6,10]. It can synthesize triacylglyceride (TAG) to over 70% of its dry cell weight, in the form of a lipid drop inside the cell [6,10], and thus has the highest accumulation level among oleaginous yeasts. Furthermore, the genomic information of *L. starkeyi* is available [11]. Its oil accumulation system reportedly activates specific biosynthesis pathways for generating TAG [12,13,14], involving many chemical reactions including some for branched structures, that finally produce TAG [15]. The TAG biosynthesis pathways are activated under nitrogen-limited conditions, and are considered to be related to cell growth, stress response and energy for survival [15]. The acyl-CoA synthesis pathway and the Kennedy pathway are important in *L. starkeyi* [16]. In the cytosol, nutrient glucose is usually converted into pyruvate, which is transported to mitochondria. In the mitochondria, pyruvate is converted into acetyl-CoA by the pyruvate dehydrogenase complex, and citrate is produced from this acetyl-CoA. Upon nitrogen source restrictions, the conversion of citrate to AKG is inhibited, and citrate accumulates in mitochondria [16,17]. This citrate is transported back to the cytosol, where it is converted into acetyl-CoA by specific enzymes in oleaginous yeasts [18]. The acetyl-CoA is converted into acyl-CoA by using malonyl-CoA molecules produced from acetyl-CoA. The cytosolic acyl-CoA is utilized in the Kennedy pathway for TAG biosynthesis [16].

Although the pathways for generating TAG have been extensively studied, the regulation of these chemical reactions has remained enigmatic, and thus we are far from the complete control of this TAG biosynthesis system. To enhance our knowledge and control the oil accumulation system in *L. starkeyi*, the transcriptional regulatory mechanisms that underlie this system must be elucidated. Gene regulatory network analysis is one of the useful methods to gain insights into transcriptional regulation. Various algorithms have been developed to infer complex gene networks from mRNA levels [19,20,21]. We developed an approach based on structural equation modeling (SEM) [22,23], which has been utilized to elucidate causal relationships in disparate fields, such as econometrics, sociology and psychology [24,25,26]. The noteworthy features of SEM are the inclusion of latent variables into the constructed model and the ability to infer the network, including cyclic structures. We previously developed a new statistical approach, based on SEM in combination with stepwise factor analysis, to infer the protein–DNA interactions for gene transcriptional control from only the gene expression profiles, in the absence of protein information [23]. Furthermore, we applied our approach to reveal the causalities within the well studied serial transcriptional regulation systems in *Saccharomyces cerevisiae*, *Caenorhabditis elegans*, *Drosophila melanogaster*, and human ES cells [27,28,29,30]. 

Here, we applied our developed SEM approach for inferring the transcriptional regulatory mechanism underlying the whole TAG biosynthesis system in *L. starkeyi*. Even though the flow of TAG biosynthesis has been studied as metabolic/biosynthesis pathways, they reveal the chemical reactions to generate TAG from glucose, rather than the transcriptional regulation of TAG production. These metabolic/biosynthesis pathways can be utilized for the dynamics equation of the mass balance of chemical compounds, and those methods were known to be useful for TAG biosynthesis control. To develop further an efficient strain for bioproduction by genetic control, we have to clarify the complicated transcriptional regulatory mechanism of TAG production, which remains unclear. In this study, we infer the structure of the transcriptional regulation of TAG biosynthesis by setting the TAG production ability as an objective variable. Since our developed SEM approach can detect the fitting scores of predicted models with the measured data, our approach is one of the powerful approaches to infer the transcriptional regulatory model for revealing the effect for result variables.

## 2. Materials and Methods

### 2.1. Gene Expression Data Processing

For our network analysis, we utilized the expression profiles of 7799 genes, including 14 mitochondrial genes, measured by DNA microarray techniques, the cell concentration data (1 × 10^8^ cells/mL), and the total amount of produced oil data (g/L) measured in several types of strains in 256 experiments. Among the 256 experiments, 184 measured 8 time points from 0 to 240 h in several types of strains, 12 measured 4 time points from 48 to 192 h in a wild-type strain or low oil productivity strain, and 60 measured 3 time points from 24 to 72 h in wild-type strains or high oil productivity strains. Together, the data points were considered to clarify when the TAG biosynthesis occurs [31]. 

First, the expression profiles of 7799 genes were described as the log_2_-value of the raw expression signals, and then transformed into Z-scores. Subsequently, we defined the oil productivity in each condition, as follows:(1)$$oil\_productivity(t)=\frac{OIL(t+1)\times 10^{3}}{Cell(t+1)}-\frac{OIL(t)\times 10^{3}}{Cell(t)}$$

Here, *OIL*(*t*) and *Cell*(*t*) are the oil production data and the cell concentration data, which were measured as the phenotypic data at every time point (*t*) in each strain, and *oil_productivity*(*t*) was calculated as the TAG productivity per cell. From a biological viewpoint, a time-gap will occur between the defined *oil_productivity*(*t*) and the state of gene expression. The effect of gene expression at time *t* can be detected as the difference in oil productivity between time *t* + 1 and time *t*. In this method, since the last time point of each time series data had no oil productivity information, these data were deleted. Finally, we utilized 210 experimental data for calculations.

### 2.2. Gene Selection

To construct the regulatory network of the TAG biosynthesis pathways, we utilized the Kyoto Encyclopedia of Genes and Genomes Database (KEGG https://www.genome.jp/kegg/pathway.html) and JGI Genome Portal Database (https://genome.jgi.doe.gov/portal/) Databases. In the KEGG Database and by prior investigations [32], the lipid biosynthesis pathway in another oleaginous yeast, *Yarrowia lipolytica*, is available. We obtained the gene information related to the TAG biosynthesis pathways in *Y. lipolytica* and searched for the homologous genes in *L. starkeyi* by using the JGI Database. Finally, 88 genes were detected as components of the TAG biosynthesis pathway in *L. starkeyi*.

### 2.3. Factor Analysis

To clarify the regulatory system of the TAG biosynthesis pathway, we determined the optimal number of subgroups that were regulated by the same system, by a factor analysis. First, the 88 selected genes and defined oil productivity were described as observed variables, and factor analyses were used to find their suitable numbers of regulatory factors.

Factor analysis is a statistical method for describing the variability among observed variables in terms of a potentially lower number of latent variables [33]. The initial assumption is that any observed variables may be related to any latent variables. Let us assume that there are p latent variables and q observed variables *x*_1_, *x*_2_, …, *x_q_*, with means *u*_1_, *u*_2_, …, *u_q_*. Note that the number p of the latent variables is always smaller than the number q of the observed variables. Each observed variable is expressed as linear combinations of p latent variables, as follows: (2)$$x_{i}-u_{i}=\alpha_{i,1}F_{1}+\alpha_{i,2}F_{2}+\cdots +\alpha_{i,q}F_{q}+\varepsilon_{i}$$
where *x_i_* is the vector of the expression levels of gene *i*, *α_i,j_* is the partial regression weight of the latent variable *F_j_*, and *ε_i_* is an independently distributed error term with zero mean and finite variance. The observed variables expressed by Equation (2) can be summarized in a matrix form: (3)X−U=AF+E

If there are *n* samples in each of the observed variables, then *X* and *U* are the (*p* × *n*) matrices composed of the observed data and their means, respectively. The partial regression coefficients of each latent variable are indicated as elements of A, the (*q* × *p*) latent interaction matrix. In matrix A, each column corresponds to a factor and each row corresponds to an observed variable, and thus each element of A indicates the strength of the regulation from each protein to each gene. The *F* matrix is the latent variable matrix, and *E* is the (*q* × *n*) error matrix. 

The variance–covariance matrix between the observed variables Σ structurized by parameters is described, as follows: (4)$$\Sigma =Var\left[X\right]=E\left[(X-U){\left(X-U\right)}^{t}\right]=A\Phi A^{t}+\Psi^{2}$$

Here, A is the factor loading matrix of latent variables, Φ is the covariance matrix among factors, and Ψ^2^ is the covariance matrix of error terms. From this structurized matrix, the values of the partial regression weight matrix A and the variances of the “errors” are estimated.

To detect the suitable number of subgroups in the TAG biosynthesis pathways, exploratory factor analysis (EFA) was performed. EFA is utilized to reveal the latent structure, by assuming that the observed data are a synthetic amount of a lower number of latent variables. In this study, EFA was executed by a principal factor method with promax rotation, which is a general method for fitting rotating factors to a hypothesized structure of latent variables. In this study, we applied EFA for dividing several subgroups of TAG biosynthesis pathway genes, so we utilized Kaiser criterion, which is one of the major criterions, as for estimating the number of factors at first. By Kaiser criterion, the number of factors is known to be overestimated. Thus, we reduced the estimated factor number one by one to confirm the suitable number of factors, which control TAG biosynthesis pathway genes. In this step, we applied a scree plot to avoid underestimating the number of factors.

### 2.4. Stepwise Network Modeling

The 88 genes detected by the factor analysis were divided into two types. One type is the genes classified into the same group with oil productivity, thus reflecting that those genes are controlled by the same transcriptional regulatory system with oil productivity. The other type is the genes classified into other groups from oil productivity, which means that those genes are controlled by a different transcriptional regulatory system than that of oil productivity. To clarify the whole regulatory system of TAG biosynthesis, we applied stepwise network modeling as follows:STEP 1:Initial model assumption of oil productivity group;STEP 2:Model optimization of oil productivity group;STEP 3:Definition of pseudo variables from subgroups;STEP 4:Initial model assumption among pseudo variables;STEP 5:Model optimization of pseudo variables.

#### 2.4.1. Initial Model Assumption

For the SEM calculations, we had to assume the initial model in each step. In this case, only one variable was defined as an objective variable, and the star model can be applied as the initial model. In STEP 1, the oil productivity was assigned as the objective variable. The genes classified into the same group with the oil productivity were assumed to be the effect variables for the oil productivity. We arranged 15 genes as the parent nodes for the oil productivity as a child node in an initial model.

The whole regulatory system for TAG biosynthesis was inferred by a pseudo variables network. Given the restrictions of the SEM calculation, it is difficult to construct the optimal network model with the selected 88 genes. We calculated the representative value of each group from the measured data of components within the group and defined one pseudo variable corresponding to one group. In the pseudo variables network, the variable including oil productivity was defined as the objective variable, and the other variables were assumed as the effect variables for the regulator of oil productivity in the initial model. Thus, 8 variables were arranged as the parent node for the one specific variable as a child node.

In the initial model, the parent nodes were assumed to be independent, and the relationships between parent nodes were not identified. Thus, the initial model was expressed by
(5)$$\left[\begin{array}{c} f\\ g \end{array}\right]=\left[\begin{array}{cc} O & O\\ \Gamma & O \end{array}\right]\left[\begin{array}{c} f\\ g \end{array}\right]+\left[\begin{array}{c} f\\ \sigma \end{array}\right].$$

Here, *f* is a vector of effect variables arranged as parent nodes, and *g* is the data of the objective variable. Since the parent nodes were assumed to be independent, the weights of the relationships between them were defined as *O* matrices. The matrix Γ is a vector representing the effectiveness of parent nodes to child nodes. The errors that affect the objective variables are denoted by *ε*.

#### 2.4.2. Network Modeling

To detect the regulatory structure of TAG biosynthesis, we previously applied our developed network modeling method based on SEM calculation [34]. We applied this method to the initial models to obtain the optimized regulatory network model. In this study, all variables in the network model were defined as observed variables, and none were defined as latent variables.

In the inferred network model, the variance–covariance matrix between the arranged *n* variables Σ(*θ*) was given by
(6)$$\Sigma \left(\theta \right)={\left(I-\Lambda \right)}^{-1}\Phi {\left(I-\Lambda \right)}^{{-1}^{\prime }}.$$

Let *I* denote the identity matrix, Λ denote the *n* × *n* matrices of the inferred parameter matrix, and Φ denote the covariance matrix of the error terms. The real covariance matrix Σ is calculated from the observed data. Each element of the model’s variance–covariance matrix Σ(*θ*) is expressed as a function of the parameter *θ*, and all parameters in Σ(*θ*) are calculated to minimize the difference from Σ by the maximum likelihood method: (7)$$F_{ML}\left(\Sigma ,\Sigma \left(\theta \right)\right)=\log\left|\Sigma \left(\theta \right)\right|-\log\left|\Sigma \right|+tr\left(\Sigma {\left(\theta \right)}^{-1}\Sigma \right)-q.$$

Here, |Σ| is the determinant of matrix Σ, *tr*(Σ) is the trace of matrix Σ, and *q* is the number of observed variables.

In the SEM calculation, the similarity between the constructed model and the actual relationships is predicted by comparing Σ(*θ*) and Σ, and the quantitative similarity can be detected as fitting scores. To evaluate the inferred model accuracy, we utilized six different fitting scores as criteria: χ^2^ values (CMIN), the goodness of fit index (GFI), the adjusted goodness of fit index (AGFI), the comparative fit index (CFI), the root mean square error of approximation (RMSEA) and the Akaike’s information criterion (AIC). These criteria indicate the qualities of model adaptation to the measured data, and they have threshold values to determine whether the model is suitable. A CMIN value higher than 0.01 was considered as a well fitted model, and GFI and CFI values above 0.90 are required for a good model fit [35,36,37]. RMSEA is one of the most popular parsimony indexes and is independent of a huge sample number. In the RMSEA criteria, values below 0.05 would represent a good model fit, even though values of 0.10 or more are considered to indicate that the constructed model is far from the actual data [38]. Furthermore, we evaluated the model fitting by AIC [39], to compare the independent model and the saturation model. We used the SEM software package SPSS AMOS 27 (IBM, Armonk, NY, USA).

To infer the optimal network structure, we applied our developed iteration algorithm for model optimization [27]. Recently, we improved our iteration algorithm to escape from a local optimal solution combined with the genetic algorithm [34]. In this algorithm, we utilized the probabilities of all edges in the inferred model, which were calculated by the inverse matrix of the Fisher information matrix of parameters, to detect the significance of each edge. The detected non-significant edges were deleted from the inferred model step by step, and all parameters were re-calculated in each step. All of the edges were detected as significant, and we utilized modification index (MI) scores, calculated from chi-square statistic, to estimate the possible relationships between variables. The regulatory relationship with the highest MI score at each step was add to the inferred model as a new edge. We applied this iteration algorithm to the observed and latent variables at first, and then the error terms were executed.

## 3. Results

### 3.1. Gene Classification by Factor Analysis

Since the TAG biosynthesis pathways are a specific system for survival under nutrient-limited conditions in *L. starkeyi*, the system was expected to be divided into some subgroups for its regulation; that is, some regulatory factors were expected to be expressed simultaneously in TAG biosynthesis-related genes. To reveal the regulatory subgroups in TAG biosynthesis, we performed exploratory factor analysis (EFA) and then applied confirmatory factor analysis (CFA) by the principal factor method with promax rotation. Since EFA is commonly used for identifying the number of factors with effects on the observed variables, we applied EFA first. After the number of factors was determined, we applied CFA to classify the observed variables strictly.

To detect the genes in the same regulatory system as the oil productivity, we executed factor analysis to 88 TAG biosynthesis-related genes and oil productivity, and thus the compiled expression profiles of 89 variables measured under 210 conditions were classified by their regulatory factors. In EFA, the Kaiser criterion was utilized to estimate the suitable number of factors. The Kaiser criterion asserts that the number of factors is the same as the number of eigen values of the covariance matrix that are greater than one, and 12 factors were extracted as regulatory factors of the 89 variables. Among the 89 variables, 88 have the highest factor loading values to nine factors, and the remaining three factors were considered to be ineffective for these variables. Furthermore, the scree plot of EFA indicated that the suitable number of extracted factors can be nine. The cumulative sum of the squared factor loadings for nine factors was 83.872%, and this means that nine factors were sufficient to explain the 89 variables. Thus, we executed CFA for 89 variables with nine factors, and they were well classified by the nine factors according to their highest factor loadings. The communality, which can clarify the percent of variance in each variable explained by nine factors, and the factor loading of each factor, are displayed in Table 1.

The communalities of the 88 TAG biosynthesis-related genes were over 0.500, and among them, 63 genes were higher than 0.800. These high communality values reflected the fact that the expression of TAG biosynthesis genes could be explained by these nine factors. On the other hand, the communality of oil productivity was 0.382. This means that the regulatory system for oil productivity was not only TAG biosynthesis genes, and other regulatory systems are involved. 

By CFA, the 89 variables were well classified into nine subgroups by their factor loadings: Group 1 had 24 genes, Group 2 had 21 genes, Group 3 had 15 genes and oil productivity, Group 4 had 9 genes, Group 5 had 5 genes, Group 6 had 6 genes, Group 7 had one gene, Group 8 had 3 genes, and Group 9 had 4 genes.

### 3.2. Oil Productivy Network: Figures, Tables and Schemes

The confirmatory factor analysis (CFA) classification of oil productivity was included in Group 3, with 15 genes that are known to be related to the reactions for TAG biosynthesis. To infer the regulatory system for oil productivity within the group, the network modeling method was applied. First, we connected the 15 genes to oil productivity as an initial model. In the initial model, the oil productivity was affected by all 15 genes.

The entire architecture of the inferred network model among 15 genes and oil productivity is shown in Figure 1a. Although the initial model has a very simple structure, the inferred model shown in Figure 1a represents a more complicated system. This inferred model included three cyclic regulations: ACL2 -> SLC1 -> PDB1 -> tid_2139 -> ACL1 -> ALC2, ACL2 -> SLC1 -> PDB1 -> ACL2, and SLC1 -> PDB1 -> tid_2139 -> SLC1. The goodness-of-fit scores, which detect the fitting level of the inferred model, are shown in Table 2. All of the indices satisfied the requirements for deciding whether the model was suited to the measured data.

To clarify the regulation of oil productivity, the strong relationships were extracted from the inferred model. The edges, which have high weight absolute values (>0.3), are displayed in Figure 1b. This model was a perfectly hierarchical structure, with no cyclic regulation. From this network, we extracted the oil productivity regulations displayed in Figure 1c. In this figure, DGA2 was reasonable as a regulator of oil productivity, since DGA2 is the gene encoding an enzyme for TAG synthesis. Interestingly, other inferred regulators of oil productivity were palmitoyl-CoA biosynthesis-related genes. In the whole network model, 46 regulatory relationships were estimated among 16 variables. Among the 46 regulatory relationships, 32 relationships were positive regulation and the remaining 14 relationships were negative. The relative strength of each association is shown as a standardized regression weight in Table 3, and all edges within the model were significant (*p* < 0.01).

### 3.3. Regulatory Network of TAG Biosynthesis

To infer the regulatory system of the TAG biosynthesis pathways, we executed our modeling method to infer the regulatory network model among the nine classified groups. By the restriction of the SEM calculation, the 89 variables should be summarized to lower numbers of variables. We classified the 89 variables into nine groups by factor analysis, and these groups were considered to reflect subgroups of the regulatory system. Thus, the inference of a regulatory network among nine groups is suitable to reveal the regulatory system of TAG biosynthesis in its entirety.

Before the application of the SEM calculation, the pseudo-data of each group should be calculated. We calculated the average values from the expression profiles of the classified genes into each group as pseudo-data. As the initial model, we assigned Group 3 as an objective variable and the other groups as effective variables, since the oil productivity variable was included in Group 3. Directed edges were connected to Group 3 from the other groups in the initial model.

The inferred network model among nine groups is displayed in Figure 2a, and the goodness-of-fit scores of each index are shown in Table 4. From these scores, all indices indicated that the inferred model is suitable for the measured data. As shown in Figure 2a, 21 regulatory relationships were estimated among nine variables, and the architecture of this inferred model was hierarchical. Among the 21 regulatory relationships, 16 were positive and 5 were negative.

The regulatory system around Group 3 extracted from the inferred model is displayed in Figure 2b. Only six groups were related to Group 3, and Groups 2 and 9 were considered to have strong regulatory relationships with Group 3. The regulations from Groups 4 and 8 to Group 3 were weak and negative, and the regulation from Group 6 to Group 3 was positive but weak. The relative strength of each association is shown as a standardized regression weight in Table 5, and all edges within the model were significant (*p* < 0.01).

## 4. Discussion

In this study, we used factor analysis to classify TAG biosynthesis genes according to their regulatory system. Group 1 includes many genes related to mitochondria or peroxisomes. Group 2 has many primary metabolism pathway-related genes, and the members of Group 3 include some genes known as regulatory factors of TAG biosynthesis. Almost all of the genes classified into Groups 4 and 5 were related to mitochondria. Group 6 included genes related to TAG biosynthesis. The cholesterol esterase TGL1 was the sole component of Group 7, Group 8 included two triosephosphate isomerase genes, and Group 9 included two pyruvate decarboxylases and one diacylglycerol acyltransferase. From the tendencies of the group members, the groups were divided into three types: the groups that are not related to or reduced TAG biosynthesis (1, 4, and 5), the groups that are related to or induced TAG biosynthesis (2, 3, and 9), and the groups that could not be characterized by their features (6, 7, and 8).

The genes classified into Group 3 included known regulatory factors for TAG biosynthesis. The induction of acyl-CoA is important for TAG biosynthesis [16]. Among the 15 genes classified into Group 3, seven genes were related to acyl-CoA synthesis. Furthermore, among the remaining eight genes, seven genes were related to the reactions from glycerol-3-P to the Kennedy Pathway. Thus, the genes within Group 3 are considered to be reasonable for encoding regulatory factors for the TAG biosynthesis, and we inferred the regulatory network of oil productivity shown in Figure 1. In this inferred network model of oil productivity, a specific feature was detected. The inferred regulatory system for oil productivity includes some cyclic structures involved in negative regulations: ACL2 -> SLC1 -| PDB1 -> tid_2139 -> ACL1 -> ALC2, ACL2 -> SLC1 -| PDB1 -> ACL2, and SLC1 -| PDB1 -> tid_2139 -| SLC1. The regulations from SLC1 to PDB1 and tid_2139 to SLC1 were negative. SLC1 is related to the conversion from lysophosphatidic acid (LPA) to phosphatidic acid (PA) within the Kennedy pathway, and PDB1 is related to the conversion from acetyl-CoA to citrate in mitochondria. Both genes are considered to be important for TAG biosynthesis. These cyclic structures within the inferred model indicate the possible existence of unknown negative feedback in the regulation of TAG biosynthesis.

The extracted regulations of oil productivity, shown in Figure 1b, included known regulators of TAG biosynthesis. In oleaginous yeast *L. starkeyi*, the expression of ACL1 and DGA2 involved in the acyl-CoA synthesis pathway and the Kennedy pathway, respectively, in the mutants with greatly elevated lipid production was increased compared to that in the wild-type strain [16,31]. Furthermore, the overexpression of ACL1 or DGA2 in *L. starkeyi* led to an increase in oil productivity (Takaku et al.; unpublished data). That is, ACL1 and DGA2 are considered to be regulatory factors and play a vital role in TAG biosynthesis in *L. starkeyi*, and our inferred network model fit well with this knowledge. Even though ACC1 is related to the conversion from acetyl-CoA to malonyl-CoA, which is important for acyl-CoA synthesis, ACC1 negatively regulated oil productivity in this model. This feature means that the excessive induction of ACC1 may have a negative effect on TAG biosynthesis. Silverman et al. examined the effect of overexpression of genes involved in TAG synthesis in the oil productivity and reported that the most drastic increase in oil productivity was the overexpression of DGA2 in oleaginous yeast *Y. lipolytica* [40]. Liu et al. also investigated the transcriptional activity of lipogenic promoters and reported that the transcriptional activity of the ACC1 promoter was decreased in the oil accumulation phase in *Y. lipolytica* [41]. These reports agree with our inferred network model, and DGA2 and ACC1 may be common regulatory factors in oleaginous yeasts.

To clarify the whole regulatory system of TAG biosynthesis, we inferred network models between the groups classified by factor analysis. Since biosynthetic systems generally have many components that are related by complicated structures, revealing the regulatory system may be useful to detect the target factors for system control. The whole regulatory system in TAG biosynthesis is shown in Figure 2a, in which the system is rather complicated, but some groups were detected as having no relation to oil productivity (Group 3). The groups with no relations to Group 3 were Groups 1 and 5, which were determined to be “the groups which are not related to or reduced TAG biosynthesis” by the factor analysis classification. The genes within the groups were related to TAG biosynthesis from a chemical reaction viewpoint, but not related to TAG biosynthesis from a regulation system viewpoint. The regulatory network model for Group 3 is shown in Figure 2b. In this figure, Groups 2 and 9 were detected as strong positive regulators for TAG biosynthesis and were determined to be “the groups that are related to or induced TAG biosynthesis” by the factor analysis. Interestingly, Group 2 has strong negative regulation to Group 8, which includes the genes related to the conversion between G3P and DHAP. DHAP is the starting compound of glycerol-3-P to the Kennedy pathway, and the conversion from DHAP to G3P is not effective for TAG biosynthesis. Actually, Group 8 has a weak but negative regulation of Group 3 in the inferred network model. Thus, the negative regulation from Group 2 to Group 8 is reasonable for Group 2, as a positive regulator of TAG biosynthesis.

## 5. Conclusions

The inferred models in this study effectively reflected the known regulatory relationships for TAG biosynthesis. Furthermore, our inferred network model detected some special features in the regulatory viewpoints, such as a negative feedback system for oil productivity and non-related genes for TAG biosynthesis regulation. This computational modeling method will help us to reveal the mechanisms underlying measured biological data.

## Figures and Tables

**Figure 1 bioengineering-07-00148-f001:**
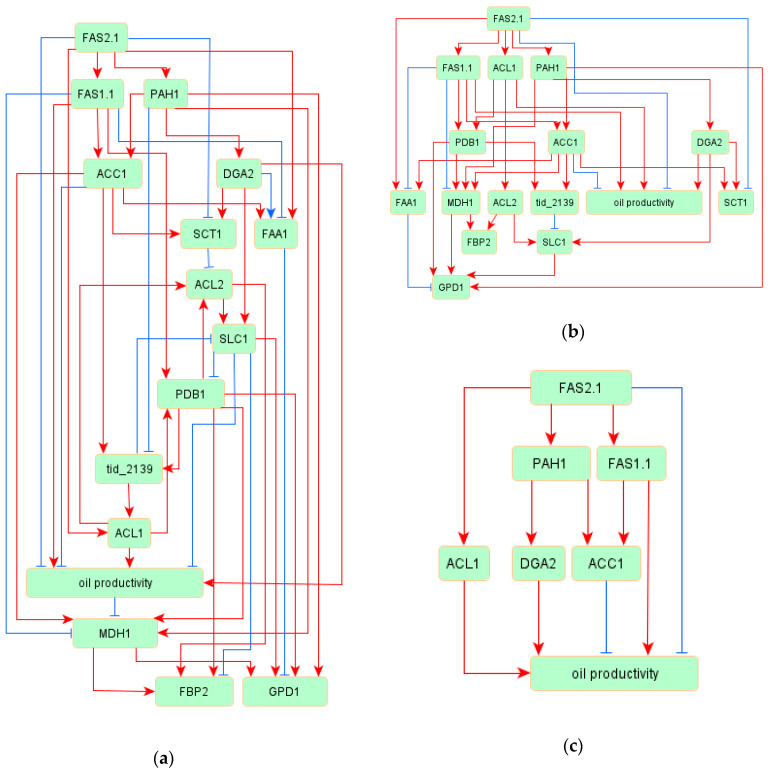
The inferred network model within the oil productivity group. Red arrows indicate positive regulations between variables, and blue arrows indicate negative regulations: (**a**) the whole inferred network model; (**b**) the extracted network model with strong regulation from the whole inferred mode; and (**c**) the regulatory model for oil productivity.

**Figure 2 bioengineering-07-00148-f002:**
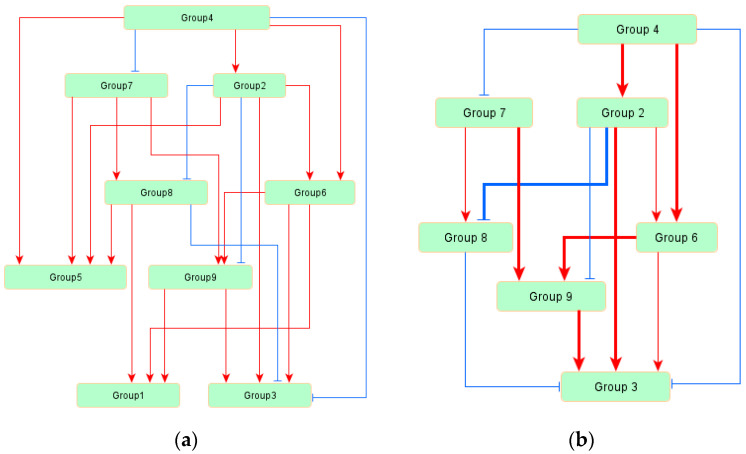
The regulatory network model for triacylglyceride (TAG) biosynthesis. Each group includes some genes as components. Red arrows indicate positive regulations, and blue arrows indicate negative regulations: (**a**) the inferred whole network model; and (**b**) the extracted regulatory network model for Group 3. Group 3 included “oil productivity” as a component, and the regulation in Group 3 means the regulation in TAG biosynthesis. Bold lines indicate strong regulations (>0.3), and thin lines indicate weak regulations (<0.3).

**Table 1 bioengineering-07-00148-t001:** Estimated factor loadings and communalities.

			Estimated Factor Loadings								
Group #	Gene	Communality	1	2	3	4	5	6	7	8	9
**1**	LACS1.1	0.951	**1.028**	0.174	−0.081	0.062	0.085	−0.034	0.001	−0.056	0.003
	POT2	0.953	**0.984**	0.125	−0.060	0.333	0.012	−0.129	0.010	0.102	0.042
	POX1	0.900	**0.966**	0.243	0.003	0.135	0.075	−0.266	0.171	0.109	0.028
	FOX2	0.960	**0.965**	−0.021	−0.061	0.097	0.037	0.044	0.119	−0.074	−0.036
	FBP1	0.907	**0.957**	0.261	−0.057	−0.067	0.354	−0.055	−0.084	−0.079	−0.249
	CIT3	0.899	**0.942**	0.107	−0.088	0.015	0.180	0.044	−0.054	0.181	0.082
	CIT2	0.913	**0.912**	0.042	0.083	−0.043	−0.086	0.065	0.044	0.131	0.016
	ALDH	0.910	**0.883**	−0.097	0.011	−0.143	0.108	0.153	−0.089	−0.080	0.074
	TGL4	0.914	**0.871**	0.082	0.046	0.109	−0.096	0.282	−0.131	−0.006	0.002
	TGL3	0.825	**0.851**	0.035	0.120	0.118	0.052	−0.192	−0.236	−0.204	0.335
	ARE2	0.932	**0.793**	−0.395	0.099	−0.072	0.027	0.008	−0.005	0.016	−0.135
	ARE1	0.822	**0.781**	−0.223	0.375	−0.045	0.066	−0.233	0.193	0.312	−0.002
	POT1	0.899	**0.720**	−0.038	−0.021	−0.085	−0.050	0.174	−0.169	0.037	0.397
	ERG13	0.905	**−0.688**	0.129	0.441	0.029	0.002	−0.048	−0.063	−0.030	0.060
	PCK1	0.782	**0.686**	−0.282	−0.082	−0.130	0.510	0.037	−0.043	0.182	0.074
	ERG10	0.915	**−0.649**	0.127	0.372	0.132	0.042	0.044	0.150	−0.012	−0.013
	ACO2	0.847	**−0.641**	0.024	−0.110	0.030	0.109	0.046	0.565	0.133	−0.077
	LAT1	0.929	**−0.605**	−0.001	0.342	−0.043	0.324	0.089	0.252	−0.102	0.135
	PDX1	0.962	**−0.595**	0.283	0.461	−0.181	0.067	−0.013	0.161	−0.015	0.048
	PYC2	0.787	**0.584**	−0.047	0.255	−0.199	0.033	0.279	−0.306	0.323	0.094
	DGK1	0.846	**−0.555**	0.299	−0.035	0.257	−0.002	−0.372	−0.115	0.100	−0.281
	AYR1	0.843	**0.551**	−0.313	0.175	0.261	−0.301	0.207	0.200	−0.330	−0.044
	PDA1	0.897	**−0.422**	0.264	0.378	−0.111	0.276	−0.016	0.210	−0.096	−0.020
	SDH2	0.874	**0.340**	0.303	−0.082	0.335	0.310	0.119	0.315	−0.196	0.153
**2**	PFK2	0.837	−0.118	**0.949**	0.001	0.028	−0.140	0.017	−0.040	0.171	0.027
	HXK2	0.882	0.007	**0.919**	0.074	0.118	−0.196	−0.311	−0.143	−0.098	0.091
	MDH2	0.908	−0.129	**0.903**	−0.045	−0.060	−0.037	0.078	0.141	0.028	−0.001
	ACS1	0.841	0.396	**0.894**	0.259	−0.063	−0.109	0.104	−0.163	0.364	0.023
	ZWF1	0.907	0.050	**0.868**	0.044	0.104	−0.197	0.097	−0.062	−0.270	−0.105
	PGM1	0.891	0.474	**0.854**	0.075	0.230	−0.330	−0.092	−0.031	−0.212	0.218
	LRO1	0.600	−0.235	**0.806**	−0.186	−0.324	−0.195	0.237	−0.251	0.135	0.112
	FUM1	0.882	−0.109	**0.794**	−0.244	−0.104	0.071	−0.196	0.362	0.044	0.135
	PGI1	0.891	0.327	**0.792**	0.092	0.195	0.217	0.025	−0.276	−0.109	−0.132
	KGD2	0.947	−0.299	**0.779**	−0.199	−0.203	0.321	0.025	−0.207	0.018	−0.031
	LSC2	0.976	−0.386	**0.761**	−0.091	0.048	0.126	−0.077	−0.087	−0.052	0.009
	CIT1	0.885	0.326	**0.743**	0.292	−0.178	0.277	0.003	0.041	−0.022	0.083
	IDH2	0.900	−0.186	**0.740**	−0.285	−0.071	0.355	−0.066	0.042	0.052	−0.011
	SDH1	0.957	0.415	**0.716**	−0.383	0.156	0.041	0.003	0.334	0.060	−0.225
	GND1	0.911	−0.024	**0.714**	0.256	−0.047	0.062	0.042	−0.073	−0.258	−0.073
	ENO1	0.919	−0.102	**0.683**	0.205	0.076	0.107	−0.058	0.094	−0.066	0.242
	PGK1	0.922	−0.293	**0.620**	0.264	−0.063	0.050	0.147	0.065	−0.151	0.016
	GUT2	0.836	0.479	**0.597**	−0.040	−0.401	−0.145	−0.233	0.582	−0.086	0.004
	CDC19	0.932	−0.147	**0.585**	0.393	0.163	−0.020	−0.020	0.055	0.056	−0.171
	HMG1	0.644	−0.211	**0.464**	0.081	0.365	−0.088	0.259	−0.123	0.323	−0.074
	LSC1	0.927	−0.415	**0.417**	0.008	0.199	0.298	−0.280	−0.124	−0.102	−0.010
**3**	PAH1	0.840	0.234	0.039	**0.931**	−0.245	−0.069	−0.010	0.006	0.034	0.084
	SCT1	0.925	0.090	−0.400	**0.924**	0.031	0.086	0.077	−0.004	−0.045	0.243
	ACC1	0.861	−0.049	−0.105	**0.923**	0.130	−0.125	0.060	0.006	0.067	0.045
	SLC1	0.921	0.121	−0.092	**0.899**	−0.238	−0.127	−0.361	0.007	−0.103	−0.139
	DGA2	0.851	0.104	−0.113	**0.882**	−0.158	−0.085	−0.209	−0.023	−0.077	0.239
	FAS1.1	0.865	−0.328	0.016	**0.833**	0.017	−0.116	−0.013	−0.081	0.012	0.038
	ACL1	0.974	−0.182	0.313	**0.793**	0.089	−0.228	−0.013	−0.014	0.042	−0.021
	GPD1	0.878	0.240	0.287	**0.728**	−0.300	0.316	−0.328	0.087	0.164	−0.037
	FAS2.1	0.865	−0.165	0.409	**0.721**	−0.073	−0.298	0.017	−0.024	0.050	0.047
	ACL2	0.971	−0.254	0.421	**0.692**	0.065	−0.222	0.028	−0.027	0.045	−0.005
	MDH1	0.882	0.458	0.270	**0.673**	−0.090	0.309	0.123	0.042	0.017	0.065
	FAA1	0.728	0.130	0.363	**0.578**	0.031	−0.349	0.149	0.226	0.050	−0.064
	FBP2	0.913	0.008	0.204	**0.568**	0.196	0.325	0.321	−0.134	0.055	−0.009
	oil productivity	0.382	−0.252	−0.105	**0.556**	−0.059	0.032	−0.062	−0.008	0.053	0.111
	tid_2139	0.645	−0.433	−0.173	**0.532**	0.430	0.167	−0.055	0.181	0.097	−0.078
	PDB1	0.891	−0.300	0.296	**0.421**	−0.013	0.324	0.086	0.103	−0.038	−0.039
**4**	YEH2	0.729	−0.095	0.117	0.153	**−0.962**	0.060	−0.147	0.223	0.054	0.117
	K_6707	0.956	0.200	0.023	0.243	**−0.931**	−0.083	0.059	0.329	0.001	−0.025
	PYC1	0.786	0.040	0.124	−0.071	**0.820**	0.004	0.169	−0.023	−0.014	−0.057
	LACS1.2	0.734	−0.308	−0.087	−0.193	**−0.799**	0.192	0.459	0.157	−0.474	−0.016
	FAS2.2	0.821	0.138	0.115	−0.186	**−0.722**	−0.304	0.151	0.100	0.188	0.025
	ACO1	0.889	0.317	0.219	−0.255	**0.624**	0.165	−0.059	0.308	−0.030	−0.029
	MAE1	0.702	0.155	−0.558	0.098	**0.564**	0.115	0.072	0.006	0.057	0.262
	SDH3	0.746	0.000	0.370	−0.109	**0.538**	−0.044	0.202	0.144	−0.129	−0.105
	tid_69043	0.708	−0.059	0.194	0.255	**0.382**	0.169	0.313	−0.046	−0.264	0.179
**5**	KGD1	0.656	0.206	−0.199	−0.016	0.070	**0.824**	0.170	−0.040	0.085	0.025
	IDH1	0.900	−0.140	0.277	−0.367	0.129	**0.664**	0.179	0.078	0.053	0.099
	TPI1	0.775	−0.052	0.061	0.201	0.214	**0.619**	0.338	−0.213	0.078	−0.022
	IDP1	0.693	0.199	0.403	−0.346	0.184	**0.605**	0.092	−0.106	0.045	0.054
	LPD1	0.819	−0.306	0.482	−0.096	−0.249	**0.515**	0.219	−0.248	0.115	−0.013
**6**	CDS1	0.760	−0.034	−0.049	0.125	−0.034	−0.243	**−0.815**	0.256	−0.028	−0.013
	ALE1	0.796	−0.056	0.244	0.219	−0.196	−0.384	**−0.648**	0.105	−0.132	−0.072
	K_291711	0.815	0.425	0.288	−0.016	−0.120	−0.036	**0.574**	0.003	−0.283	−0.160
	GAP1	0.712	−0.184	−0.212	0.297	0.243	0.133	**0.544**	0.067	−0.041	0.197
	EMI2	0.766	0.223	0.272	0.293	0.283	0.066	**0.478**	−0.089	−0.084	0.143
	SHH4	0.882	0.431	−0.099	−0.111	0.417	0.347	**0.439**	0.071	0.147	−0.339
**7**	TGL1	0.611	0.157	0.135	−0.052	0.209	0.151	0.201	**−0.828**	0.219	0.192
**8**	TPI2.2	0.730	0.377	−0.038	0.033	−0.237	0.273	0.093	−0.269	**0.616**	−0.082
	TPI2.1	0.792	0.414	−0.428	0.231	−0.008	0.122	−0.064	−0.079	**0.603**	0.004
	FAS1.2	0.560	−0.460	0.009	−0.305	−0.059	0.066	0.059	−0.114	**0.478**	−0.008
**9**	PDC1	0.851	0.167	0.030	0.426	−0.068	0.179	0.246	−0.272	0.022	**0.665**
	DGA1	0.684	0.357	0.044	0.427	−0.183	−0.063	−0.103	0.099	−0.235	**0.554**
	PDC2	0.846	0.043	−0.404	0.160	−0.179	0.022	0.021	−0.318	0.209	**0.518**
	SOL3	0.768	−0.153	0.166	0.298	0.461	0.089	−0.134	−0.300	0.028	**0.497**

Note: The highest value of factor loadings is colored in red.

**Table 2 bioengineering-07-00148-t002:** Goodness-of-fit scores for the inferred model.

	CMIN (*P*)	GFI	AGFI	CFI	RMSEA	AIC
Estimated model	0.032	0.958	0.892	0.996	0.043	239.68
Saturated model		1		1		272
Independent model	0	0.151	0.038	0	0.447	5166.72

Note: the probability from χ^2^ values (CMIN(*P*)), the goodness of fit index (GFI), the adjusted goodness of fit index (AGFI), the comparative fit index (CFI), the root mean square error of approximation (RMSEA) and the Akaike’s information criterion (AIC).

**Table 3 bioengineering-07-00148-t003:** Estimated regulations and regression weights.

Source	Target	Standardized Regression Weight	*p* Values
FAS2.1	FAS1.1	0.894	***
FAS2.1	PAH1	0.695	***
FAS1.1	ACC1	0.668	***
PAH1	DGA2	0.827	***
PAH1	ACC1	0.339	***
ACC1	SCT1	1.148	***
DGA2	SCT1	0.306	***
FAS2.1	SCT1	−0.555	***
FAS2.1	ACL1	0.765	***
DGA2	SLC1	0.572	***
ACC1	tid_2139	0.443	***
SCT1	ACL2	−0.112	***
PAH1	tid_2139	−0.193	0.002
FAS1.1	PDB1	0.436	***
ACL1	oil productivity	0.487	***
ACC1	oil productivity	−0.36	0.007
FAS1.1	oil productivity	0.687	***
FAS2.1	oil productivity	−0.479	0.002
SLC1	oil productivity	−0.254	0.007
DGA2	oil productivity	0.488	***
FAS2.1	FAA1	1.117	***
PAH1	MDH1	0.619	***
PDB1	MDH1	0.615	***
ACC1	FAA1	0.786	***
oil productivity	MDH1	−0.107	0.005
FAS1.1	MDH1	−0.802	***
FAS1.1	FAA1	−0.98	***
DGA2	FAA1	−0.175	***
ACC1	MDH1	0.608	***
MDH1	FBP2	0.486	***
MDH1	GPD1	0.327	***
SLC1	GPD1	0.3	***
PDB1	GPD1	0.422	***
PDB1	FBP2	0.254	***
SLC1	FBP2	−0.182	***
ACL2	FBP2	0.3	***
FAA1	GPD1	−0.398	***
PAH1	GPD1	0.362	***
ACL1	ACL2	0.991	***
PDB1	tid_2139	0.504	***
ACL1	PDB1	0.429	***
PDB1	ACL2	0.081	***
tid_2139	SLC1	−0.342	***
tid_2139	ACL1	0.26	***
SLC1	PDB1	−0.225	***
ACL2	SLC1	0.545	***

Asterisks (***) within the table indicate values less than 0.001. Source and target mean parent node and child node in the model, respectively. The regression weight means the strength of the regulation from source to target.

**Table 4 bioengineering-07-00148-t004:** Goodness-of-fit scores for the TAG biosynthesis regulatory model.

	CMIN (*P*)	GFI	AGFI	CFI	RMSEA	AIC
Estimated model	0.027	0.973	0.913	0.981	0.064	87.91
Saturated model		1		1		90
Independent model	0	0.592	0.49	0	0.289	683.446

**Table 5 bioengineering-07-00148-t005:** Estimated regulations and regression weights of the TAG biosynthesis regulatory model.

Source	Target	Standardized Regression Weight	*p* Values
Group4	Group2	0.353	***
Group4	Group7	−0.285	***
Group4	Group6	0.472	***
Group2	Group6	0.249	***
Group2	Group8	−0.383	***
Group7	Group8	0.178	0.004
Group6	Group9	0.364	***
Group2	Group9	−0.177	0.009
Group7	Group9	0.362	***
Group6	Group1	0.549	***
Group2	Group5	0.557	***
Group7	Group5	0.304	***
Group4	Group5	0.24	***
Group9	Group1	0.157	0.006
Group2	Group3	0.462	***
Group8	Group1	0.233	***
Group9	Group3	0.476	***
Group8	Group5	0.215	***
Group8	Group3	−0.152	0.001
Group6	Group3	0.241	***
Group4	Group3	−0.189	***

Note: Asterisks (***) within the table indicate values less than 0.001. Source and target mean parent node and child node in the model, respectively. The regression weight means the strength of regulation from source to target.

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
