# Peer review of "Inference of Regulatory System for TAG Biosynthesis in Lipomyces starkeyi"

_bioengineering, 2020, doi:10.3390/bioengineering7040148_

Round 1

Reviewer 1 Report

#### General comments:

In this manuscript, the authors develop a mathematical model for the oil
production of Lipomyces starkeyi. I have to admit that I am not familiar with
the type of model that is developed. Models are used to make predictions that
can be tested against measurements. The manuscript contain no information and
no comments on this matter.

On line 357 the authors say "DGA2 and ACL1 have been reported as regulatory
factors of TAG biosynthesis [16], and our inferred network model fit well with
this knowledge." This part should be more specific. How does no model agree and
are there examples of the opposite?

The discussion feels a little bit like a rehash of the results section. I think
the authors should discuss the potential usefulness of the model more
specifically and also in principle how it might be tested.

Also, can the authors motivate on the method for determining the number of
factors? The Kaiser criteria and the Scree plot are but two of a range of
methods.

The level of the English is very good and I could only find a few suggestions
of style (see below).

#### Specific comments:

The numbers below are line number in the manuscript.

----
23
"From the inferred oil productivity regulatory model, the well
characterized genes DGA1 and ACL1 were detected as regulatory factors for oil productivity."

Consider changing to:

"From the inferred oil productivity regulatory model, the well
characterized genes DGA1 and ACL1 were detected as regulatory factors."

----
41
"It produces edible oil with a fatty acid composition similar to that of palm oils, with a
reasonable cost [6-9]."

Consider changing to:

It produces edible oil with a fatty acid composition similar to that of palm oils, at a
reasonable cost [6-9].
----
93-94
Equation 1 should be edited for clarity without 45 degree slashes
----
229
In the Kaiser criterion, the number of factors is the same as the number of eigen values of
the covariance matrix that are greater than one, and 12 factors were extracted as regulatory factors
of the 89 variables.

Consider changing to:

The Kaiser criterion asserts that ...

Or something to that effect.
----
242
This means that regulatory system for oil productivity
was not only TAG biosynthesis, and other regulatory systems are involved.

This seems to be an error? Should it be "TAG biosynthesis genes" instead of
"TAG biosynthesis".

Author Response

We wish to express our strong appreciation to the reviewers for their insightful comments on our paper. We feel the comments have helped us significantly improve the paper.

Comment1

Models are used to make predictions that can be tested against measurements. The manuscript contain no information and no comments on this matter.

Response1

We thank the reviewer for this comment. We agree that this point requires clarification of our model, and have added the following text to the Introduction (p. 2, lines 77-89):

“Here, we applied our developed SEM approach for inferring the transcriptional regulatory mechanism underlying the whole TAG biosynthesis system in L. starkeyi. Even though the flow of TAG biosynthesis has been studied as metabolic/biosynthesis pathways, they reveal the chemical reactions to generate TAG from glucose, rather than the transcriptional regulation of TAG production. These metabolic/biosynthesis pathways can be utilized for dynamics equation for the mass balance of chemical compounds, and those methods were known to be useful for TAG biosynthesis control. To develop further efficient strain for bioproduction by genetic control, we have to clarify the complicated transcriptional regulatory mechanism of TAG production, which is remained unclear. In this study, we infer the structure of transcriptional regulation of TAG biosynthesis by setting the TAG production ability as an objective variable. Since our developed SEM approach can be detect the fitting scores of predicted models with the measured data, our approach is one of the powerful approaches to infer the transcriptional regulatory model for revealing the effect for result variables.”

Comment2

On line 357 the authors say "DGA2 and ACL1 have been reported as regulatory factors of TAG biosynthesis [16], and our inferred network model fit well with this knowledge." This part should be more specific. How does no model agree and are there examples of the opposite?

Response2

We strongly appreciate the reviewer's comment on this point.. We agree that this point requires clarification, and have added the following text to the Discussion (p. 12, lines 374-379):

“In oleaginous yeast L. starkeyi, the expression of ACL1 and DGA2 involved in the acyl-CoA synthesis pathway and the Kennedy pathway, respectively, in the mutants with greatly elevated lipid production was increased compared to that in the wild-type strain [16, 31]. Furthermore, overexpression of ACL1 or DGA2 in L. starkeyi led to increase in oil productivity (Takaku et al.; unpublished data). That is, ACL1 and DGA2 are considered to be regulatory factors and play a vital role in TAG biosynthesis in L. starkeyi,”

Comment3

The discussion feels a little bit like a rehash of the results section. I think the authors should discuss the potential usefulness of the model more specifically and also in principle how it might be tested.

Response3

We strongly appreciate the reviewer's comment on this point. We agree that this point requires clarification, and have added the following text to the Discussion (p.12, lines 384-389) and References (p.15, lines 514-519):

“Silverman et al. examined the effect of overexpression of genes involved in TAG synthesis in the oil productivity and reported that the most drastic increase in oil productivity was overexpression of DGA2 in oleaginous yeast Y. lipolytica [40]. Liu et al. also investigated the transcriptional activity of lipogenic promoters and reported that the transcriptional activity of ACC1 promoter was decreased in oil accumulation phase in Y. lipolytica [41]. These reports agree with our inferred network model, and DGA2 and ACC1 may be common regulatory factors in oleaginous yeasts.”

References:

  1. Silverman, A. M.; Qiao, K.; Xu, P.; Stephanopoulos, G. Functional overexpression and characterization of lipogenesis-related genes in the oleaginous yeast Yarrowia lipolytica. Appl. Microbiol. Biotechnol. 2016, 100(8), 3781-3798.
  2. Liu, H.; Marsafari, M.; Deng, L.; Xu, P. Understanding lipogenesis by dynamically profiling transcriptional activity of lipogenic promoters in Yarrowia lipolytica. Appl. Microbiol. Biotechnol. 2019, 103(7), 3167-3179.

Comment4

Also, can the authors motivate on the method for determining the number of factors? The Kaiser criteria and the Scree plot are but two of a range of methods.

Response4

We wish to express our deep appreciation to the reviewer for his insightful comment on this point. To make this point clearer, we have added the following to the Materials and Methods 2.2 (p.4, lines 152-158):

“In this study, we applied EFA for dividing several subgroups of clustering the TAG biosynthesis pathway genes, so we utilized Kaiser criterion, which is one of the a major criterions, as for estimating the number of factors at first. By Kaiser criterion, the number of factors is known to be overestimated. Thus, we reduce the estimated factor number one by one to confirm the suitable number of factors, which control TAG biosynthesis pathway genes. In this step, we applied a scree plot to avoid underestimating the number of factors.”

Specific Comment5-1

Line 23 "From the inferred oil productivity regulatory model, the well characterized genes DGA1 and ACL1 were detected as regulatory factors for oil productivity."
Consider changing to:
"From the inferred oil productivity regulatory model, the well characterized genes DGA1 and ACL1 were detected as regulatory factors."

Response to Specific Comment5-1

We thank the reviewer for this comment. In accordance with the reviewer's comment, we have changed Line 23 as follows:

“From the inferred oil productivity regulatory model, the well characterized genes DGA1 and ACL1 were detected as regulatory factors.”

Specific Comment5-2

Line 41 "It produces edible oil with a fatty acid composition similar to that of palm oils, with a reasonable cost [6-9]."

Consider changing to:

It produces edible oil with a fatty acid composition similar to that of palm oils, at a reasonable cost [6-9].

Response to Specific Comment5-2

We thank the reviewer for this comment. In accordance with the reviewer's comment, we have changed Line 42-43 as follows:

“It produces edible oil with a fatty acid composition similar to that of palm oils, at a reasonable cost [6-9].”

Specific Comment5-3

Line 93-94 Equation 1 should be edited for clarity without 45 degree slashes

Response to Specific Comment5-3

We thank the reviewer for this comment. In accordance with the reviewer's comment, we have changed the equation without 45 degree slashes.

Specific Comment5-4

Line 229 “In the Kaiser criterion, the number of factors is the same as the number of eigen values of the covariance matrix that are greater than one, and 12 factors were extracted as regulatory factors of the 89 variables.”

Consider changing to:

The Kaiser criterion asserts that ...

Or something to that effect.

Response to Specific Comment5-4

We thank the reviewer for this comment. In accordance with the reviewer's comment, we have changed Line 246 as follows:

“The Kaiser criterion asserts that In the Kaiser criterion, the number of factors is the same as the number of eigen values of the covariance matrix that are greater than one, and 12 factors were extracted as regulatory factors of the 89 variables.”

Specific Comment5-5

Line 242 “This means that regulatory system for oil productivity was not only TAG biosynthesis, and other regulatory systems are involved.”

This seems to be an error? Should it be "TAG biosynthesis genes" instead of "TAG biosynthesis".

Response to Specific Comment5-5

We thank the reviewer for this comment. In accordance with the reviewer's comment, we have changed Line 260 as follows:

“This means that regulatory system for oil productivity was not only TAG biosynthesis genes, and other regulatory systems are involved.”

Reviewer 2 Report

Structural equation modeling (SEM) is an important tool to study and analyze gene regulatory network. The authors here developed an SEM-based model to elucidate the protein-DNA interactions for gene transcriptional control of TAG synthesis in the lipogenesis organism L. starkeyi. Reveling the complicated gene regulatory network is important for us to infer critical metabolic engineering targets that could be modified to improve TAG synthesis.

Before this manuscript is accepted, the authors are suggested to address a number of concerns.

(1) In the method part, I didn't see the structure-based kinetic/mass balance equations that govern TAG productivity or TAG titer. Are the SEM model is generic and not constrained by the detailed enzyme-regulatory reaction networks?

(2) In recent years, people have been increasingly used the ensemble model to study gene regulatory network. Please describe what is the difference between the SEM approach (the model/algorithm in this paper) and the ensemble approach.

(3) The authors used Y. lipolytica's lipid synthesis as the reference point. Lipogenesis in Y. lipolytica is induced and controlled by the amount of nitrogen present to the media. Nutrient factors, Carbon/Nitrogen ratio has been reported to regulat the IDH activity of TCA cycle through AMP deaminase, which converts AMP to IMP. Experimental results have demonstrated that energy charge and NADPH controls lipid content in Y. lipolytica. Have these factors or constraints incorporated to the SEM model?

(4) DNA microarry data is used to infer the gene regulatory network. At protein level, ACC1 phosphorylation has been reported to strongly related with lipogenesis. It seems the regulatory node of ACC1 is not included in this model, please explain.

(5) A number of recent studies have tested gene overexpression targets and promoter activity that are related with lipogenesis activity in Y. lipolytica. Please have a brief discussion of the results reported in (a) Silverman et al, Functional overexpression and characterization of lipogenesis-related genes in the oleaginous yeast Yarrowia lipolytica, AMB 2016; and (b) Liu et al,Understanding lipogenesis by dynamically profiling transcriptional activity of lipogenic promoters in Yarrowia lipolytica, AMB 2019. For example, how the experimental data reported in these two work support the findings of the SEM model reported in this work. This could be a way to make the current work be validated or discussed from the reference strain or metabolic network used in this study.

Author Response

We wish to express our strong appreciation to the reviewers for their insightful comments on our paper. We feel the comments have helped us significantly improve the paper. In particular, we wish to acknowledge their highly valuable comment 5.

Comment1

In the method part, I didn't see the structure-based kinetic/mass balance equations that govern TAG productivity or TAG titer. Are the SEM model is generic and not constrained by the detailed enzyme-regulatory reaction networks?

Response1

We strongly appreciate the reviewer's comment on this point. We agree with the reviewer that kinetic/mass balance equations are useful. However, in this study we try to clear the structure of transcriptional regulation by expression profiles. We agree that this point requires clarification of our research purpose, and have added the following text to the Introduction (p. 2, lines 77-85):

“Here, we applied our developed SEM approach for inferring the transcriptional regulatory mechanism underlying the whole TAG biosynthesis system in L. starkeyi. Even though the flow of TAG biosynthesis has been studied as metabolic/biosynthesis pathways, they reveal the chemical reactions to generate TAG from glucose, rather than the transcriptional regulation of TAG production. These metabolic/biosynthesis pathways can be utilized for dynamics equation for the mass balance of chemical compounds, and those methods were known to be useful for TAG biosynthesis control. To develop further efficient strain for bioproduction by genetic control, we have to clarify the complicated transcriptional regulatory mechanism of TAG production, which is remained unclear.”

Comment2

In recent years, people have been increasingly used the ensemble model to study gene regulatory network. Please describe what is the difference between the SEM approach (the model/algorithm in this paper) and the ensemble approach.

Response2

We strongly appreciate the reviewer's comment on this point. We agree that this point requires clarification, and have added the following text to the Introduction (p. 2, lines 85-89):

“In this study, we infer the structure of transcriptional regulation of TAG biosynthesis by setting the TAG production ability as an objective variable. Since our developed SEM approach can be detect the fitting scores of predicted models with the measured data, our approach is one of the powerful approaches to infer the transcriptional regulatory model for revealing the effect for result variables.”

Comment3

The authors used Y. lipolytica's lipid synthesis as the reference point. Lipogenesis in Y. lipolytica is induced and controlled by the amount of nitrogen present to the media. Nutrient factors, Carbon/Nitrogen ratio has been reported to regulat the IDH activity of TCA cycle through AMP deaminase, which converts AMP to IMP. Experimental results have demonstrated that energy charge and NADPH controls lipid content in Y. lipolytica. Have these factors or constraints incorporated to the SEM model?

Response3

We appreciate the reviewer's comment on this point. We agree that additional information on Nutrient factors as the reviewer suggested would be valuable. Regrettably, however, because of our study was designed to focus inferring the transcriptional regulatory mechanism underlying the whole TAG biosynthesis system in L. starkeyi. Thus, posttranslational regulation such as the regulation of IDH activity of TCA cycle through AMP deaminase is not incorporated into SEM approach. But, the reviewers suggestion is very valuable for us, and we want to take the reviewers point as a consideration for future study.

Comment4

DNA microarry data is used to infer the gene regulatory network. At protein level, ACC1 phosphorylation has been reported to strongly related with lipogenesis. It seems the regulatory node of ACC1 is not included in this model, please explain.

Response4

We thank the reviewer for this comment. We agree with the reviewer that ACC1 phosphorylation related with lipogenesis. However, our study was designed for inferring the structure of transcriptional regulation for TAG biosynthesis pathway genes. Thus, posttranslational regulation such as the regulation of ACC1 by phosphorylation is not incorporated into this SEM approach unfortunately. But, this reviewer’s comment is very helpful for us, and we consider to take this reviewer’s point as a consideration for our future studies. To clear this point in this manuscript, we have added the following text to the Introduction (p.2: lines 77-78)

“Here, we applied our developed SEM approach for inferring the transcriptional regulatory mechanism underlying the whole TAG biosynthesis regulatory system in L. starkeyi.

Comment5

A number of recent studies have tested gene overexpression targets and promoter activity that are related with lipogenesis activity in Y. lipolytica. Please have a brief discussion of the results reported in (a) Silverman et al, Functional overexpression and characterization of lipogenesis-related genes in the oleaginous yeast Yarrowia lipolytica, AMB 2016; and (b) Liu et al,Understanding lipogenesis by dynamically profiling transcriptional activity of lipogenic promoters in Yarrowia lipolytica, AMB 2019. For example, how the experimental data reported in these two work support the findings of the SEM model reported in this work. This could be a way to make the current work be validated or discussed from the reference strain or metabolic network used in this study.

Response5

We wish to express our deep appreciation to the reviewer for his insightful comment on this point. We agree that this point requires clarification, and have added the following text to the Discussion (p.12, lines 384-389) and References (p.15, lines 514-519):

“Silverman et al. examined the effect of overexpression of genes involved in TAG synthesis in the oil productivity and reported that the most drastic increase in oil productivity was overexpression of DGA2 in oleaginous yeast Y. lipolytica [40]. Liu et al. also investigated the transcriptional activity of lipogenic promoters and reported that the transcriptional activity of ACC1 promoter was decreased in oil accumulation phase in Y. lipolytica [41]. These reports agree with our inferred network model, and DGA2 and ACC1 may be common regulatory factors in oleaginous yeasts.”

References:

  1. Silverman, A. M.; Qiao, K.; Xu, P.; Stephanopoulos, G. Functional overexpression and characterization of lipogenesis-related genes in the oleaginous yeast Yarrowia lipolytica. Appl. Microbiol. Biotechnol. 2016, 100(8), 3781-3798.
  2. Liu, H.; Marsafari, M.; Deng, L.; Xu, P. Understanding lipogenesis by dynamically profiling transcriptional activity of lipogenic promoters in Yarrowia lipolytica. Appl. Microbiol. Biotechnol. 2019, 103(7), 3167-3179.
